# Racial Disparities in Periprosthetic Joint Infections after Primary Total Joint Arthroplasty: A Retrospective Study

**DOI:** 10.3390/antibiotics12111629

**Published:** 2023-11-16

**Authors:** Jodian A. Pinkney, Joshua B. Davis, Jamie E. Collins, Fatma M. Shebl, Matthew P. Jamison, Jose I. Acosta Julbe, Laura M. Bogart, Bisola O. Ojikutu, Antonia F. Chen, Sandra B. Nelson

**Affiliations:** 1Massachusetts General Hospital, Boston, MA 02114, USA; fshebl@mgh.harvard.edu (F.M.S.);; 2Harvard Medical School, Boston, MA 02115, USA; jcollins13@bwh.harvard.edu (J.E.C.);; 3Brigham and Women’s Hospital, Boston, MA 02115, USA; jdavis83@bwh.harvard.edu (J.B.D.); mpjjamison@gmail.com (M.P.J.); jacostajulbe@bwh.harvard.edu (J.I.A.J.); 4RAND Corporation, Santa Monica, CA 90401, USA; 5Charles R. Drew University of Medicine and Science, Los Angeles, CA 90059, USA; 6Boston Public Health Commission, Boston, MA 02118, USA

**Keywords:** periprosthetic joint infection, racial disparities, arthroplasty, health equity

## Abstract

In the United States, racial disparities have been observed in complications following total joint arthroplasty (TJA), including readmissions and mortality. It is unclear whether such disparities also exist for periprosthetic joint infection (PJI). The clinical data registry of a large New England hospital system was used to identify patients who underwent TJA between January 2018 and December 2021. The comorbidities were evaluated using the Elixhauser Comorbidity Index (ECI). We used Poisson regression to assess the relationship between PJI and race by estimating cumulative incidence ratios (cIRs) and 95% confidence intervals (CIs). We adjusted for age and sex and examined whether ECI was a mediator using structural equation modeling. The final analytic dataset included 10,018 TJAs in 9681 individuals [mean age (SD) 69 (10)]. The majority (96.5%) of the TJAs were performed in non-Hispanic (NH) White individuals. The incidence of PJI was higher among NH Black individuals (3.1%) compared with NH White individuals (1.6%) [adjusted cIR = 2.12, 95%CI = 1.16–3.89; *p* = 0.015]. Comorbidities significantly mediated the association between race and PJI, accounting for 26% of the total effect of race on PJI incidence. Interventions that increase access to high-quality treatments for comorbidities before and after TJA may reduce racial disparities in PJI.

## 1. Introduction

Periprosthetic joint infection (PJI) is defined as an infection involving a joint prosthesis and its surrounding tissues following total joint arthroplasty (TJA) [1]. In the United States (US), the incidence of PJI in the general population ranges from 0.5 to 2.3% after total knee arthroplasty (TKA) and 0.5 to 2.1% after total hip arthroplasty (THA) [2,3,4]. The treatment of PJI is associated with high morbidity, mortality, and staggering costs [1,4,5,6,7]. In 2018, the estimated inpatient cost of treating PJIs that developed after TKA and THA was $608 million and $399 million, respectively, including costs related to antibiotic therapy and revision arthroplasties [4]. Individuals with PJI also experience a lower quality of life compared to the general population, with quality-adjusted life year (QALY) measurements ranging from 0.28 to 0.83, where 1 represents perfect health and 0 represents death [8,9,10].

The extent of racial and ethnic disparities in PJI is unknown because data for distinct racial and ethnic groups in PJI incidence have been aggregated with other post-operative TJA complications [11]. For instance, an observational study of 83,887 patients in Connecticut found that Black adults were 68% more likely to have a 30-day readmission following TJA for complications including non-orthopaedic infections, deep surgical site infections, and surgical site hematomas compared to White adults [12]. While these results suggest that the incidence of PJI is higher among Black individuals, these data have not been explicitly reported. Other factors that suggest that Black individuals may have a higher incidence of PJI include racial disparities in most of the underlying risk factors for PJI. For instance, in the US, racial disparities have been observed in access to dental care [13], high-quality obesity management [14], smoking cessation resources [15], and high-quality diabetes care [16], which are all factors known to increase the risk of PJI.

Acquiring data on the incidence of PJIs stratified by race is the initial step in determining whether racial disparities exist for this medical condition. It would also serve as a foundation for developing effective interventions and policies to mitigate racial disparities [17]. Baseline epidemiological data stratified by race also plays a vital role in informing modeling and cost-effectiveness studies [4,9,11].

The primary objective of this study is to compare the incidence of PJI between non-Hispanic (NH) White and NH Black individuals. We also examined whether the presence of comorbidities or income inequalities mediated the relationship between race and PJI. Throughout this paper, we use the word race to refer to the social construct based on physical appearance often utilized as a proxy for capturing information about unmeasured variables influencing health outcomes, including exposure to racism and its consequential effects [18,19,20,21,22].

## 2. Methods

We used the Research Patient Data Registry (RPDR), which captures data from 10 hospitals in the New England region, to identify a convenience sample of patients who underwent primary TKA or primary THA between January 2018 and December 2021 [23]. We extracted the data in July 2023 so that the estimated duration of follow-up for each arthroplasty ranged from 19 to 42 months. This study was approved by the Massachusetts General Hospital Institutional Review Board (Protocol Number: 2023P000586). We used Current Procedural Terminology (CPT^®^) codes to identify primary TKAs and THAs (Appendix A). International Classification of Diseases Tenth Revision (ICD-10) codes were used to identify PJI following these TJAs (Appendix A). We used the Agency for Healthcare Quality and Research (AHRQ) Elixhauser Comorbidity Index (ECI) software tool to generate ECI readmission index scores for all patients based on the comorbidities present at the time of their TJA [24,25,26]. The AHRQ ECI readmission index score uses 26 of the 33 comorbidities included in the original ECI [24,26]. Additional demographic information, including age and sex, was extracted from the data registry. Manual chart reviews for 100 arthroplasties were performed by two study staff members to verify the integrity and quality of the data extracted from the data registry.

### 2.1. Measures

#### 2.1.1. Primary Outcome

The primary outcome was a diagnosis of PJI following primary TKA or THA. This was collected as a binary variable (i.e., yes vs. no) for each type of arthroplasty.

#### 2.1.2. Primary Exposure

The RPDR captures race and ethnicity data from the enterprise master patient index (EMPI) database, a large database used to ensure that each patient in a hospital data system is represented only once with consistent demographic identifiers for all systems. Race was categorized using the following subgroups: White, Black, Asian, American Indian or Alaska Native, Hawaiian, other, unknown, and declined. Ethnicity was extracted as a binary variable (Hispanic or non-Hispanic). We subsequently recoded race/ethnicity as a binary variable: NH White or NH Black. Based on the low numbers, we excluded individuals in the other racial/ethnic categories (Figure 1). In our manual review of the first 100 patients in the Epic™ electronic medical record system, we found 100% concordance for racial categories between Epic™ and the RPDR; however, there was an 18% discordance for ethnic categories between Epic™ and the RPDR. This was attributed to a change that occurred in 2021, when patients were able to manually enter their own ethnicity data in their medical records.

#### 2.1.3. Potential Confounders and Mediators

Age at the time of the primary TJA was extracted as a continuous variable. Legal sex (e.g., the sex designated on an individual’s driver’s license or U.S. state identification) was extracted as a binary variable (male or female). The comorbidity data were extracted as binary variables for each of the 26 comorbidities in the AHRQ ECI software tool [25,26]. The AHRQ ECI software tool sums weighted points from each comorbidity diagnosis based on ICD-10-CM codes to generate a composite ECI readmission index score that determines the risk of 30-day readmission. Negative weights indicate a lower risk of 30-day readmission, while positive weights indicate a higher-risk, with higher scores indicating greater readmission risk [26]. Information regarding median household income was based on zip code data from the 2021 American Community Survey [27].

### 2.2. Inclusion/Exclusion Criteria

The initial dataset included 12,071 TJAs with 202 PJIs. We excluded 1143 TJAs because ethnicity was categorized as Hispanic, declined, unknown, or missing. We then excluded 543 TJAs because race was not categorized as Black or White (Figure 1). Finally, we excluded 367 individuals with missing comorbidity data, leaving 10,018 arthroplasties in 9681 individuals in the final analytic dataset (Figure 1). Overall, 32/202 (16%) PJIs and 2021/11,869 (17%) non-PJIs were excluded.

### 2.3. Statistical Analyses

Descriptive statistics are presented as frequencies and percentages for categorical variables and means and standard deviations (SD) for continuous variables. Bivariate associations between race or PJI and a set of a priori sociodemographic variables were determined using Poisson regression for PJI and logistic regression with generalized estimating equations for race. We used Poisson regression models to assess the association between race and PJI using adjusted cumulative incidence ratios (cIRs) and 95% confidence intervals (95% CIs). We adjusted for age and sex in model 2 and age, sex, and comorbidities in model 3. We examined the interaction between race and sex regarding the incidence of PJI. Because the interaction was not significant, it was not included in the final multivariable Poisson regression model.

Certain comorbidities are known to increase PJI risk; for example, PJI risk doubles in patients with obesity or diabetes [28], while substance abuse may increase PJI risk 9-fold [29]. Based on known racial inequalities in the distribution of wealth and comorbidities in the US [30], both of which can impact an individual’s risk of developing PJI, we performed mediation analyses to examine whether income and the composite AHRQ ECI scores mediated the relationship between race and PJI. We used the structural equation model (SEM) framework to examine the theoretical model in which we proposed that the relationship between race and PJI was partially mediated by the ECI and income and confounded by age and sex (Figure 2A). We used Mplus 8.6 to employ the SEM analysis [31]. Using the cross-product method, we calculated the indirect effect coefficient (mediated effect) as the product of the coefficient of the path from race to the mediators (ECI and income) and the coefficient of the path from the mediators to PJI. We also estimated the direct effect coefficient between race and PJI and the total effect coefficient by summing the direct and indirect effect coefficients. We used the standardized root mean square residual (SRMR), comparative fit index (CFI), root mean square error of approximation (RMSEA), and the coefficient significance to guide the modification of the model to identify the final model [32].

## 3. Results

### 3.1. Descriptive Statistics

In the final analytic dataset, there were 10,018 TJAs from 9681 individuals. There were 9344 NH White individuals contributing 9664 (96.5%) TJAs and 340 NH Black individuals contributing 354 (3.5%) TJAs (Table 1). The NH Black individuals were significantly younger (mean age 65 years; SD ± 12) than the NH White individuals (mean age 69 years; SD ± 10) at the time of the primary TJA (*p* < 0.001). Females represented a higher proportion of the NH Black individuals compared to the NH White individuals (63.8% vs. 52.8%; *p* < 0.001). The majority of TJAs were TKAs in both the NH White and NH Black individuals (54.0% and 54.5%, respectively). The median annual household income was lower among NH Black individuals (mean $82,736; SD ± 31,718) compared to NH White individuals (mean $107,809; SD ± 39,380). Nearly all (25/26) comorbidities incorporated in the composite AHRQ ECI readmissions index score occurred more frequently in NH Black individuals compared to NH White individuals, except for metastatic cancer (Table 1). The composite AHRQ ECI readmissions index score was significantly higher among NH Black individuals (mean 27.5; SD ± 26.9) compared to NH White individuals (mean 18.2; SD ± 22.6), indicating higher readmission risk.

Within each racial group, the females were older than the males at the time of the primary TJA (Appendix A). NH Black females had the highest prevalence of obesity (55.8%) and uncomplicated diabetes (34.5%) and the highest mean ECI score among all four race by sex intersection groups: NH Black females (28.4), NH Black males (25.9), NH White females (18.3), and NH White males (18.2). NH Black males had the highest prevalence of substance abuse (17.2%) (Appendix A).

### 3.2. Bivariate Associations

Overall, the incidence of PJI was 17 times higher following TKA compared to THA [crude cIR = 17.26, 95%CI = 8.50–35.05; *p* < 0.0001] (Appendix A). All PJIs that developed following the primary TJA among NH Black individuals occurred following TKA and 95% of PJIs among NH White individuals occurred following TKA (Appendix A). Overall, individuals with obesity were 50% more likely to be diagnosed with PJI compared with individuals who were not obese. Similarly, individuals with uncomplicated diabetes were 59% more likely to develop PJI compared with individuals without diabetes. Individuals who had a diagnosis of substance abuse were twice as likely to be diagnosed with PJI compared with individuals without a diagnosis of substance abuse [crude cIR = 2.45, 95%CI = 1.61–3.75; *p* < 0.0001] (Appendix A).

### 3.3. Primary Outcome

The incidence of PJI was higher among NH Black individuals (3.1%) compared to NH White individuals (1.7%) in the unadjusted model [crude cIR = 1.89, 95%CI = 1.04–3.44; *p* = 0.038] (Appendix A). This association remained significant after adjusting for age and sex [adjusted cIR = 2.12, 95%CI = 1.16–3.89; *p* = 0.015] but was no longer significant after adjusting for age, sex, and comorbidities [adjusted cIR = 1.65, 95%CI = 0.90–3.03; *p* = 0.104].

Within each racial group, the incidence of PJI varied by sex, with NH Black males having the highest incidence and NH White females having the lowest incidence (Figure 3). However, there was no significant interaction between race and sex on the incidence rate of PJI; the association between race and PJI was similar for males and females. In the models adjusted for age, NH Black males were at a 1.94 times increased risk of PJI compared to NH White males, and NH Black females were at a 2.3 times increased risk of PJI compared to NH White females.

### 3.4. Mediation Analyses

We modified the theoretical model until we achieved the model that best fits the data. The final model had an excellent fit (CFI = 1, SRMR = 0.001, RMSEA (90% CI) 0 (0.0, 0.0)). After modifying the model, only the mediated effect via ECI was retained, and age and income variables were dropped from the model. Males were more likely to experience PJI than females (β(SE) = 0.24(0.06), *p* < 0.0001), and higher AHRQ ECI readmissions index scores were associated with higher PJI probability (β(SE) = 0.008(0.001), *p* < 0.0001). The AHRQ ECI readmissions index score was significantly higher in NH Black compared to NH White individuals (β(SE) = 9.26(1.05), *p* < 0.0001 (Table 2). We also detected a significant total effect (β(SE) = 0.30(0.14), *p* = 0.030), a significant indirect effect via ECI (β(SE) = 0.08(0.01), *p* < 0.0001), and a non-significant direct effect via ECI (β(SE) = 0.22(0.14), *p* = 0.104) (Table 2, Figure 2B). Approximately 26% of the effect of race on PJI (i.e., the indirect effect from race to ECI to PJI divided by the total effect) was mediated by ECI (β(SE) = 0.26(0.12), *p* = 0.036) (Table 2).

## 4. Discussion

Whether the incidence of PJI following TJA differs across diverse racial groups represents a critical gap in the literature [2,3,4]. Understanding whether there are racial disparities in PJI is an important step toward achieving health equity by the year 2030, a national goal established by the U.S. Office of Disease Prevention and Health Promotion [33,34]. In this large secondary analysis, three key findings emerged: (1) the incidence of PJI in NH Black individuals was twice that of NH White individuals after adjusting for age and sex, (2) the presence of comorbidities significantly mediated the relationship between race and PJI and explained 26% of the total effect of race on the incidence of PJI, and (3) at the intersection of race and sex, NH Black males had the highest incidence of PJI (3.7 per 100 population), while NH White females had the lowest incidence (1.2 per 100 population). Similar to previous studies, we found that TKA posed a greater risk of PJI compared with THA in both racial groups [35,36]; 100% of PJIs among NH Black individuals and 95% of PJIs among NH White individuals occurred following TKA. We found that several comorbidities, including obesity, diabetes, and substance abuse, were associated with a higher likelihood of PJI and were more prevalent among NH Black individuals compared with NH White individuals; higher ECI scores were also associated with increased incidence of PJI.

### 4.1. The Incidence of PJI in NH Black Individuals Was Twice That of NH White Individuals after Adjusting for Age and Sex

This study adds to the growing body of literature on racial disparities in post-operative outcomes following TJA. A large Connecticut-based retrospective analysis of all-payer data on 63,036 TKAs and 39,474 THAs performed between 2005 and 2015 found that NH Black patients were 68% more likely to have 30-day readmission compared with NH White patients; however, disparities improved from 2009 to 2015 [12]. Similarly, a retrospective analysis of national Medicare claims data on 1,483,221 TJAs performed from 2013 to 2018 found that NH Black adults were 38% more likely to have 90-day readmission compared to NH White patients in 2013. However, readmission rates among NH Black patients decreased by 24% from 2013 to 2018, narrowing racial disparities [11]. In both studies, PJI was listed among several reasons for readmission, but PJI incidence was not specifically reported [11,12]. In both studies, disparities narrowed over time. The improvement in disparities was postulated to be due to a heightened national emphasis on value-based care [12].

### 4.2. The Presence of Comorbidities Significantly Mediated the Relationship between Race and PJI and Explained 26% of the Total Effect of Race on the Incidence of PJI

Using mediation analysis, our study found that an unequal distribution of comorbidities between racial groups partially explained the effect of race on the incidence of PJI. Consequently, it is important to understand the reasons for the higher prevalence of comorbidities in NH Black individuals compared to their NH White counterparts, as it forms the basis for designing effective mitigating interventions. Specifically, our study found a higher prevalence of obesity, diabetes, and substance abuse among NH Black individuals compared to their NH White counterparts [14,16,37,38]. The mean ECI scores for NH Black individuals with and without PJI were 48.3 and 26.8, respectively, compared to 32.1 and 18.0, respectively, for NH White individuals. The differences in disease burden between racial groups have been attributed to multiple factors that affect both disease prevention and treatment. For example, structural racism, including redlining, has led to inequities in housing opportunity and neighborhood disinvestment, which are associated with racial disparities in health outcomes [39,40,41,42]. Neighborhoods with larger populations of NH Black individuals reported higher rates of environmental toxins, lack of green spaces, obesogenic food environments, lower density of primary care and subspecialty providers, and higher levels of exposure to discrimination and stress, which triggers the biological cascade that promotes obesity, diabetes, and substance use [38,41,43]. For instance, a study examining the density of primary care providers in Philadelphia neighborhoods found that 32% of regions with ≥80% NH Black residents had few primary care providers compared to 6.1% of those regions with ≤20% NH Black residents [43]. In Massachusetts, data from the 2021 Behavioral Risk Factor Surveillance System (BRFSS) showed that 9.7% of NH Black adults reported the lack of a primary care provider compared with 5.6% of NH White adults [44]. NH Black adults also had disproportionately lower access to high-quality anti-diabetic and anti-obesity medications, endocrinologists that manage obesity and diabetes, nutritionists, bariatric surgery, and weight loss programs [14,16,38,45]. Data from the 2003–2019 Medical Expenditure Panel Survey (MEPS) showed that NH Black adults consistently had the lowest utilization of medications used in the management of diabetes and obesity, such as glucagon-like peptide-1 receptor agonists (GLP-1 RA), among all racial groups. NH Black adults also had decreased access to medications for opioid use disorder (MOUD) and were less likely to be referred to an OUD treatment facility following nonfatal overdose compared to their White counterparts [46,47]. Given the crucial role that the optimization of these three comorbidities plays in preventing PJI [48], multi-sector policies and programs that address the underlying causes of disparities should be prioritized. This includes the pressing need for neighborhood investment, care coordination, and pharmacoequity [45,49].

### 4.3. At the Intersection of Race and Sex, NH Black Males Had the Highest Incidence of PJI

In our study, at the intersection of race and sex, we found that NH Black males had the highest incidence rate of PJI (3.7 per 100 population), while the incidence rates among Black females and White males were similar (2.8 and 2.2 per 100 population, respectively). The finding that males had a higher incidence of PJI compared to females aligns with the findings from a retrospective study that used the Healthcare Cost and Utilization Project Nationwide Inpatient Sample (HCUP-NIS) from 2003 to 2011. Among 6,123,637 patients undergoing TJA, males undergoing TJA were younger and had lower obesity rates than females, yet had higher ECI scores and were 40% more likely to develop a surgical site infection [50]. Males also incurred slightly higher hospital costs than females ($15,322 vs. $15,011; *p* < 0.001), despite having a slightly shorter length of stay (3.5 vs. 3.7 days; *p* < 0.001) [50]. In addition, males had a 60% higher mortality rate and a 60% higher rate of sepsis following TJA compared to females [50]. Therefore, NH Black males fall at the intersection of poor outcomes related to both race and sex, a concept described as intersectionality [51,52]. This highlights the importance of disaggregated data in identifying particularly vulnerable racial subgroups of individuals that require prioritization of mitigating efforts and resources.

### 4.4. Limitations

The 18% discordance in ethnic category data between the RPDR and the manual chart review could have affected the accuracy of estimates of racial disparities because misclassified individuals could be incorrectly excluded or included in the analysis. There were also low numbers of other racial and ethnic groups leading to their exclusion from the analysis. We were unable to obtain information on other important variables such as interpretable insurance payer data, individual level income, education level, national origin, injection drug use, tobacco use, and medical mistrust, recognizing these as important factors in determining health outcomes and access. The PJI incidence rates reported in this manuscript were crude, as the total duration of follow-up for each arthroplasty was unavailable, and therefore precise incidence rates could not be calculated. Some PJIs may have been missed if PJI care was provided at a different hospital.

### 4.5. Future Directions

Quality improvement (QI) projects targeting the prevention of hospital readmissions after TJA, such as the Surgical Care Improvement Project (HCIP) and the Comprehensive Care for Joint Replacement Program (CJR), have been associated with lower readmission rates for NH Black patients and the narrowing of racial disparities [11,12,53]. By reducing disparities, national QI projects focused on the provision of value-based care also have the potential to lower healthcare costs [4,54]. Therefore, understanding the decision-making process for hospitals that accept or decline to participate in national QI projects will be an important area of study in the future. Ensuring that accurate and interpretable payer data can be extracted from large registries used for secondary analyses will also be important to better understand the underlying drivers of disparities [11].

Mixed-methods studies are needed to further understand the unique barriers and needs of people who develop PJI. Results from such studies could be used to inform multi-sector interventions and further refine programs and policies that address PJI and the management of comorbidities. The expansion of community- and institutional-level educational interventions, such as the Enhanced Preoperative Education Pathways (EPrEP) [55], could also significantly lower PJI disparities in the future.

Other areas for future study include the exploration of racial and ethnic disparities in immunological markers and pharmacokinetics among the individuals who develop PJI.

## 5. Conclusions

NH Black individuals are twice as likely to develop PJI following TJA compared to their NH White counterparts, and NH Black males are at particularly high risk. Comorbidities such as obesity, diabetes, and substance abuse partially explain this disparity. Efforts to eliminate racial disparities in PJI incidence will therefore require long-term community investment that promotes equity and increases access to primary care providers, subspecialists, and pharmacies in predominantly NH Black neighborhoods. Mixed methods research along with multi-sector policies and expansion of effective QI programs are needed to improve TJA outcomes and reduce healthcare costs.

## Figures and Tables

**Figure 1 antibiotics-12-01629-f001:**
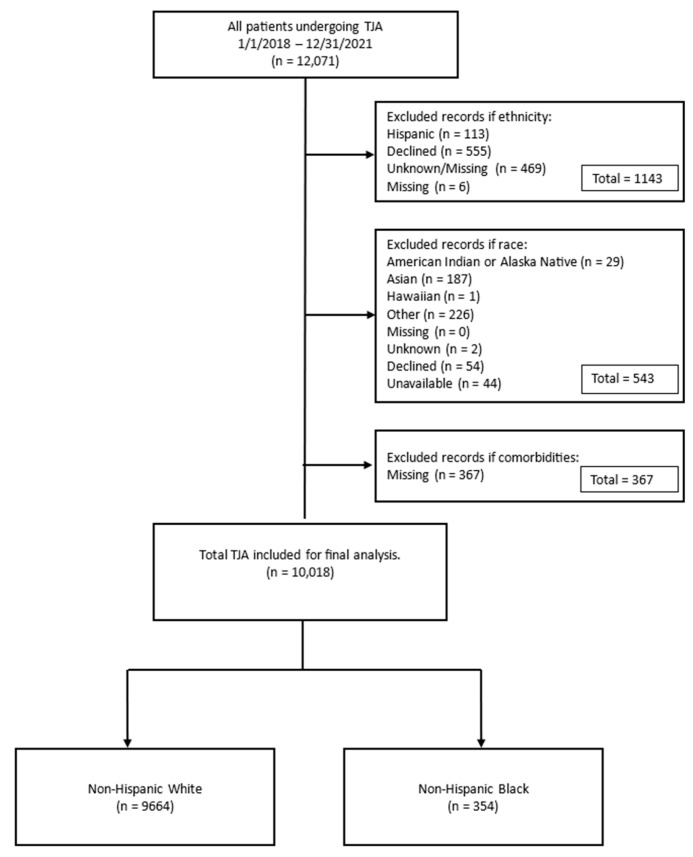
Flowchart depicting the process of exclusion used to determine the final analytic sample. TJA—total joint arthroplasty.

**Figure 2 antibiotics-12-01629-f002:**
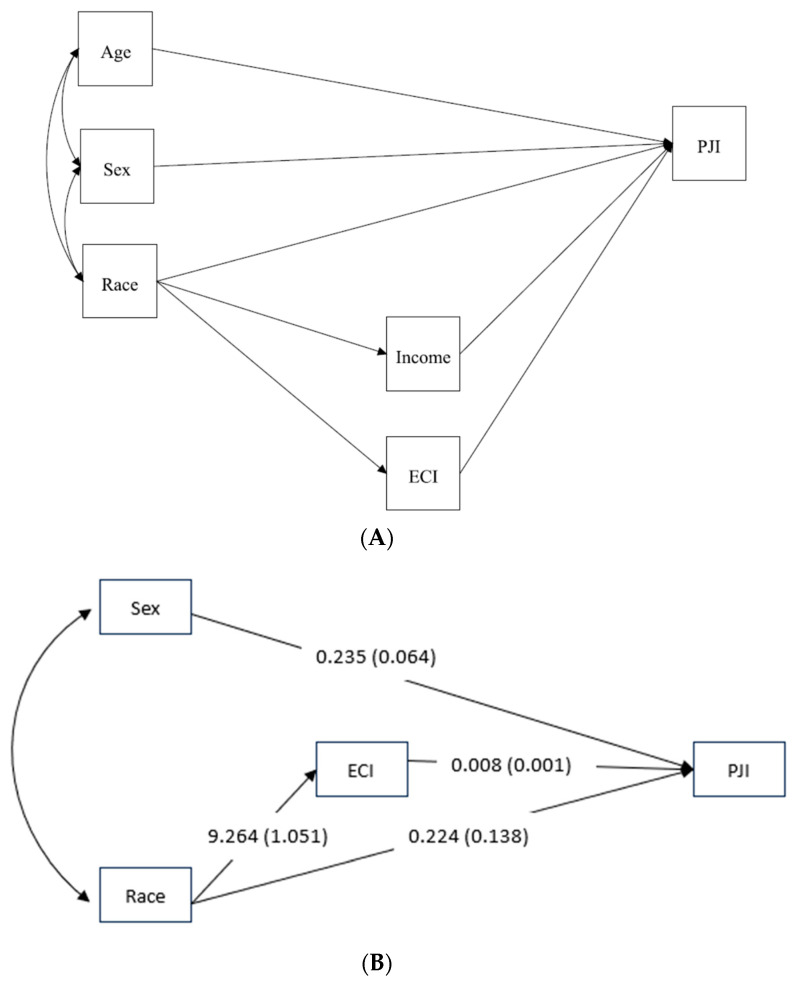
(**A**) Theoretical model of the association between race and periprosthetic joint infection. (**B**) Final model of the association between race and periprosthetic joint infection. ECI—Elixhauser Comorbidity Index; PJI—periprosthetic joint infection.

**Figure 3 antibiotics-12-01629-f003:**
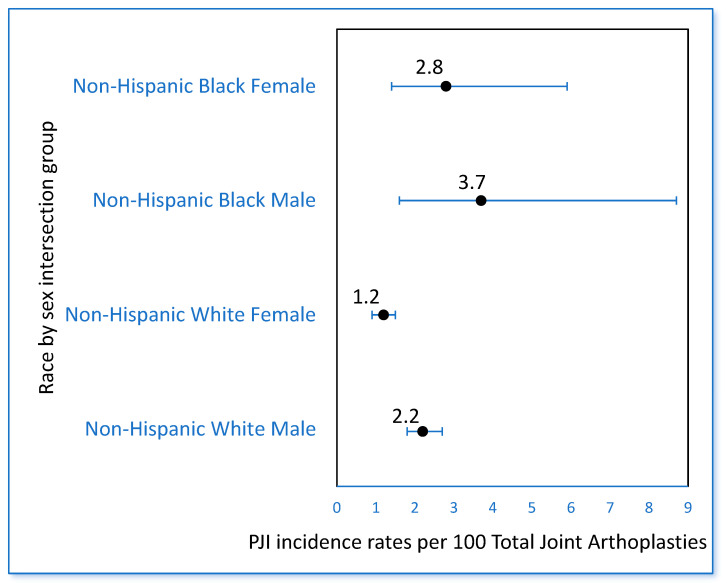
Cumulative incidence rates of periprosthetic joint infection (PJI) in four race by sex intersection groups.

**Table 1 antibiotics-12-01629-t001:** Baseline characteristics of the study population stratified by race.

Characteristic	NH White (n = 9664) (no. %)	NH Black (n = 354) (no. %)
Age—years, mean (±SD)	69 (±10)	65 (±12)
Sex		
Female	5102 (52.8)	226 (63.8)
Male	4562 (47.2)	128 (36.2)
Median Household Income †—mean (±SD)	107,809 (±39,380)	82,736 (±31,718)
Type of TJA *		
TKA *	5215 (54.0)	193 (54.5)
THA *	4449 (46.0)	161 (45.5)
Comorbidities		
Obesity	2814 (29.1)	168 (47.5)
Hypertension, uncomplicated	6.066 (62.8)	279 (78.8)
Hypertension, complicated	1153 (11.9)	85 (24.0)
Chronic pulmonary disease	2255 (23.3)	120 (33.9)
Diabetes, uncomplicated	1331 (13.8)	114 (32.2)
Diabetes, complicated	854 (8.8)	73 (20.6)
Renal failure	1.040 (10.8)	79 (22.3)
Liver disease	1015 (10.5)	59 (16.7)
HIV/AIDS	21 (0.2)	3 (0.8)
Metastatic cancer	540 (5.6)	13 (3.9)
Rheumatological disorders	1261 (13.0)	70 (19.8)
Blood loss anemia	272 (2.8)	26 (7.3)
Deficiency anemia	831 (8.6)	61 (17.2)
Alcohol abuse	506 (5.2)	24 (6.8)
Substance abuse	585 (6.1)	44 (12.4)
Psychoses	81 (0.8)	8 (2.3)
Depression	2582 (26.7)	118 (33.3)
Elixhauser Comorbidity Index—mean (±SD)	18.2 (±22.6)	27.5 (±26.9)

* Abbreviations: NH—non-Hispanic; TJA—Total Joint Arthroplasty; THA—Total Hip Arthroplasty; TKA—Total Knee Arthroplasty. † Median household income was calculated using zip code data.

**Table 2 antibiotics-12-01629-t002:** Unstandardized results of PJI by race mediation analysis.

	Estimate	S.E.	Est./S.E.	*p*-Value (Two-Tailed)
PJI regressed on				
Sex	0.235	0.064	3.691	<0.0001
Race	0.224	0.138	1.626	0.104
ECI	0.008	0.001	7.140	<0.0001
ECI regressed on				
Race	9.264	1.051	8.815	<0.0001
Effects from race to ECI to PJI				
Total	0.301	0.138	2.173	0.030
Indirect	0.077	0.014	5.559	<0.0001
Direct	0.224	0.138	1.626	0.104

ECI = Elixhauser Comorbidity Index; PJI = Periprosthetic joint infection; S.E. = Standard Error.

## Data Availability

The authors confirm that the data supporting the findings of this study are available in the article and Appendix A.

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
