# Peer review of "Racial Disparities in Periprosthetic Joint Infections after Primary Total Joint Arthroplasty: A Retrospective Study"

_antibiotics, 2023, doi:10.3390/antibiotics12111629_

Round 1

Reviewer 1 Report

Comments and Suggestions for Authors

The scientific article titled "Racial Disparities in Periprosthetic Joint Infection After Total Joint Arthroplasty" sheds light on a critical issue within the field of orthopedics and healthcare disparities. The study explores the incidence of periprosthetic joint infection (PJI) in the context of total joint arthroplasty (TJA) among different racial groups in the United States. This research addresses a significant gap in the literature by focusing on a specific complication following TJA and highlights the potential racial disparities associated with it. The study's findings have important implications for improving healthcare equity and the overall quality of care for patients undergoing TJA.

One of the innovative aspects of this study is its approach to understanding and quantifying racial disparities in PJI. By utilizing a large clinical data registry and employing Poisson regression analysis, the researchers were able to examine the relationship between race, comorbidities, and PJI incidence comprehensively. Furthermore, the study introduces the concept of mediation analysis to investigate whether comorbidities mediate the association between race and PJI. This innovative statistical technique provides insights into the underlying mechanisms contributing to racial disparities in PJI and suggests potential areas for intervention.

The study also highlights the importance of intersectionality by considering the impact of both race and sex on PJI incidence. The finding that NH Black males have the highest incidence of PJI underscores the need for a nuanced understanding of disparities that considers multiple intersecting factors.

In conclusion, this scientific article contributes valuable insights into the field of orthopedics and healthcare disparities. It underscores the significance of addressing racial disparities in PJI incidence and provides a foundation for future research and interventions aimed at reducing these disparities. Additionally, the use of mediation analysis and consideration of intersectionality are innovative approaches that enhance the depth of understanding in this area of study. The findings of this research have the potential to inform policies and interventions that improve healthcare equity for patients undergoing TJA.

Author Response

Thank you.

Reviewer 2 Report

Comments and Suggestions for Authors

According to research report, Black individuals have a higher risk of inflammation and disease occurrence compared to White individuals. The author should evaluate the relationship between the characteristics of Black individuals that cause that higher risk.

1. What is the main question addressed by the research?

Ans: Yes, it is
2. Do you consider the topic original or relevant in the field? Does it
address a specific gap in the field? Ans: The information already known is disparities in disease or risk between white and black individuals (ethnicity). For novelty, the author should explore further what differences between ethnicity affect disease occurrence, such as immunology markers, pharmacokinetics profile, or other patient characteristics.
3. What does it add to the subject area compared with other published
material?
Ans: The information already known is disparities in disease or risk between white and black individuals (ethnicity). For novelty, the author should explore further what differences between ethnicity affect disease occurrence, such as immunology markers, pharmacokinetics profile, or other patient characteristics.   4. What specific improvements should the authors consider regarding the
methodology? What further controls should be considered?
Ans: The author needs to add information about the minimum sample that represents the population, also their decision for choosing Hispanic participants.   5. Are the conclusions consistent with the evidence and arguments presented
and do they address the main question posed? Ans: Yes, it is
6. Are the references appropriate? Ans: Yes, it is
7. Please include any additional comments on the tables and figures. Ans: The table is appropriate, but further analysis such as multivariable logistic regression is needed to predict factors associated with disease occurrence in black individuals. Comments on the Quality of English Language

Moderate editing of English language required.

Reviewer 3 Report

Comments and Suggestions for Authors

Dear Authors

A very interesting study. Kindly discuss better all variables summarized in the Table 1. It is better to provide subtitles in the discussion section for each discussed variable. kindly summarize the total studies that highlight similar findings. 
